# Efficient Isolation and Functional Characterization of Niche Cells from Human Corneal Limbus

**DOI:** 10.3390/ijms23052750

**Published:** 2022-03-02

**Authors:** Naresh Polisetti, Lyne Sharaf, Ursula Schlötzer-Schrehardt, Günther Schlunck, Thomas Reinhard

**Affiliations:** 1Eye Center, Medical Center-Faculty of Medicine, University of Freiburg, Killianstrasse 5, 79106 Freiburg, Germany; lyne.sharaf@uniklinik-freiburg.de (L.S.); guenther.schlunck@uniklinik-freiburg.de (G.S.); thomas.reinhard@uniklinik-freiburg.de (T.R.); 2Department of Ophthalmology, University Hospital Erlangen, Friedrich-Alexander-University of Erlangen-Nürnberg, Schwabachanlage 6, 91054 Erlangen, Germany; ursula.schloetzer-schrehardt@uk-erlangen.de

**Keywords:** limbal stem cells, limbal niche cells, mesenchymal stem cells, melanocytes, limbal epithelial progenitor cells, corneal tissue engineering, 3D co-cultures, limbal stem cell niche

## Abstract

The fate decision of limbal epithelial progenitor cells (LEPC) at the human corneal limbus is determined by the surrounding microenvironment with limbal niche cells (LNC) as one of its essential components. Research on freshly isolated LNC which mainly include limbal mesenchymal stromal cells (LMSC) and limbal melanocytes (LM) has been hampered by a lack of efficient protocols to isolate and purify these cells. We devised a protocol for rapid retrieval of pure LMSC, LM and LEPC populations by collagenase digestion of limbal tissue and subsequent fluorescence-activated cell sorting (FACS) using antibodies against CD90 and CD117. The sorted cells were characterized by immunophenotyping and functional assays. The effects of LMSC and LM on LEPC were studied in 3D co-cultures and LEPC differentiation status was assessed by immunohistochemistry. Enzymatic digestion and flow sorting yielded pure populations of LMSC (CD117^−^CD90^+^), LM (CD117^+^CD90^−^), and LEPC (CD117^−^CD90^−^). The LMSC exhibited self-renewal capacity (55.0 ± 4.6 population doublings), expressed mesenchymal stem cell markers (CD73, CD90, CD105, and CD44), and transdifferentiated to adipocytes, osteocytes, or chondrocytes. The LM exhibited self-renewal capacity and sustained melanin production. The sorted LEPC expressed epithelial progenitor markers (CK14, CK19, and CK15) and showed a colony-forming ability. Co-cultivation of LMSC and LM with LEPC resulted in a 4–5-layered stratified epithelium and supported the preservation of a LEPC phenotype, as reflected by increased p63^+^ and Ki67^+^ cells and decreased CK12^+^ cells compared with LEPC monocultures. A highly efficient isolation of pure LM, LMSC, and LEPC populations from a single preparation may allow for direct transcriptomic and proteomic profiling as well as functional studies on native unpassaged LNC, which can be considered as proper equivalents of LNC in vivo. The developed biomimetic 3D co-culture method could provide an experimental model for investigating the functional role of LNC in the limbal stem cell niche.

## 1. Introduction

Limbal epithelial stem/progenitor cells (LEPC) are located at a specific anatomic location referred to as the limbal stem cell niche and regulate homeostasis of the corneal epithelium (Gonzalez G et al., 2018). The limbal stem cell niche is characterized by a specific extracellular matrix (ECM) composition, limbal vasculature, and surrounding limbal niche cells (LNCs) [1,2,3]. ECM composition influences the fate of LEPC by adhesion receptors and physical interactions, whereas surrounding LNC provide diverse molecular signals as cues for LEPC maintenance and differentiation [3,4]. LNCs mainly comprise intraepithelial melanocytes (limbal melanocytes, LM) and subepithelial stromal cells (limbal mesenchymal stromal cells, LMSC), which have been shown to support the corneal epithelial regeneration during wound healing and maintenance of LEPC phenotype both in vitro and in vivo [3,5,6,7,8,9]. Thus, the co-cultivation of LEPC with LMSC/LM could represent an improved strategy to generate cell transplants for patients suffering from limbal stem cell deficiency [10,11]. In addition, both LMSC and LM were shown to have potent immunomodulatory, anti-inflammatory and anti-angiogenic properties, making them attractive tools for clinical use [9,12,13,14]. To date, research on freshly isolated LMSC or LM, which may closely resemble the LNC in vivo, has been hampered by the lack of an efficient and fast protocol for isolating these cells as pure populations.

Previously, LEPC and surrounding LMSC and LM have been isolated either by enzymatic digestion of limbal tissue (dispase or collagenase or in combination) or by explant culture of limbal tissue followed by enrichment using cell type-specific media [15,16,17,18,19,20,21,22]. The main disadvantage of these methods is the contamination by other cell types, namely the presence of fibroblasts in both LEPC and LM populations [5,21,23]. Collagenase digestion of limbal tissue results in cell clusters consisting of 20% niche cells (mainly stromal cells and melanocytes) and 80% epithelial cells, whereas a combination of dispase and collagenase yields clusters composed of approximately 95% niche cells [18]. Very few studies on LM isolation have been reported and most of them rely on differential cytotoxic effects of G418 (geneticin) to prevent rapid overgrowth by epithelial cells and fibroblasts [5,23]. Recently, we have developed a protocol for isolation of melanocytes using CD117 as a selection marker [24]. However, this protocol also requires two rounds of flow sorting to eliminate contaminating stromal cells in order to get a pure LM population. Thus, current protocols for LMSC and LM purification require culturing of at least one cell passage in order to eliminate contaminating cells. Earlier studies reported that pure populations of LEPC were obtained by flow sorting based on expression of CD200 [25], stage-specific embryonic antigen-4 [26], a combination of integrin alpha 6 and CD71 [27], ATP-binding cassette sub-family 5 [28], N (neural)-cadherin [29], or Hoechst dye efflux ability [30]. However, there has been no report so far for the instant isolation of pure populations of LEPC, LMSC, and LMs from a single preparation with maximum yield.

Therefore, the aim of this study was to establish a technique for instantaneous retrieval of pure LEPC, LMSC, and LM populations from organ cultured corneal samples by means of fluorescence-activated cell sorting (FACS) using CD117 (as a surface marker of melanocytes) [24] and CD90 (as a surface marker of stromal cells) [21] as selection markers. The essence of this new approach is that stromal cells are simultaneously extracted from the total limbal cell population, which minimizes the risk of stromal contamination. The phenotypic profiles, growth and functional characteristics of sorted cells were analyzed. Furthermore, the role of LMSC and LM on LEPC phenotype was studied using a 3D co-culture system.

## 2. Results

### 2.1. Localization of Limbal Niche Cells In Situ

Immunohistochemical staining of limbal tissue (Figure 1) revealed that melanocytes (Melan-A^+^ (red) vimentin^+^(cyan) cells, arrow heads) were in close contact with clusters of cytokeratin (CK)15^+^; CK14^+^, CK19^+^ (green) LEPC cells, whereas sub-epithelial stromal cells (vimentin^+^ cells, cyan, arrows) were in close association with basal limbal epithelial cells and not with more superficial CK3^+^ cells (green) (dashed line represents the basement membrane (BM)) (Figure 1A). Double immunostaining confirmed the co-localization of CD90 (green) and vimentin (cyan) in sub-epithelial stromal cells (white arrows), which were in close association with basal layers of limbal epithelium (dotted line represents the BM), as well as blood vessels of the limbal stroma (Figure 1B, yellow arrows). Limbal sections showed the co-localization of CD117 (green) and Melan A (red) in the melanocytes (arrow heads) at the basal layer of limbal epithelium (Figure 1B).

Immunostaining of cultured limbal clusters (Figure 1C) derived from collagenase digestion showed the expression of keratins (pan-cytokeratin (PCK), green) and vimentin (cyan) in epithelial cells; Melan-A (red) and vimentin (cyan) expression in melanocytes (arrow heads), which are interspersed between the epithelial cells and vimentin expression in stromal cells (Figure 1C, arrows). Double immunostaining confirms the presence of CD90^+^ stromal cells (green, arrow) at the edge of clusters and in between the epithelial cells (epithelial (E)-cadherin, red, dashed line represents the edge of the cluster), whereas Melan-A^+^ melanocytes were always interspersed between epithelial cells (Figure 1C, red, arrow heads).

### 2.2. Flow Sorting of Limbal Niche Cells

The limbal cell suspensions were gated on forward scatter (FSC-A) and side scatter (SSC-A) to select cells of interest based on size and granularity (Figure 2(Ai)). To remove doublets or clumps, side scatter area vs. width was used to enrich single cells (Figure 2(Aii)) followed by dead cell exclusion using 4′,6-diamidino-2-phenylindole (DAPI) (Figure 2(Aiii)). Then, gates were set based on the isotype controls to select CD117^+^, CD90^+^, and CD90^−^CD117^−^ cells (Figure 2(Aiv)). Limbal-cluster derived cell suspensions from donor corneal samples provided a yield of 1.2 ± 0.3% of CD117^+^ cells and 2.4 ± 1.0% of CD90^+^ cells (Figure 2(Ai)). The total limbal population from isotype controls (Fug, 2(Aiv)) and the CD117^−^CD90^−^ (Figure 2(Av)) population were also retrieved. The number of CD117^+^ (353–1045), CD90^+^ (154–900), and CD90^−^CD117^−^ cells (68,850–288,111) per limbus varied from sample to sample (Figure 2B). The cultured CD90^+^CD117^−^ cells showed spindle-shaped morphology, elongated with prominent nucleolus (days 3 and 7, Figure 2C) and exhibited PCK^−^/Vimentin^+^/Melan-A^−^ phenotype on immunostaining (day 10, Figure 2D), a characteristic feature of LMSCs. CD90^−^CD117^+^ cells showed large, flattened, smooth bodies with multiple dendrites (days 3 and 7, Figure 2C) and stained for PCK^−^/Vimentin^+^/Melan-A^+^ (day 10, Figure 2D), a characteristic feature of melanocytes. CD90^−^CD117^−^ cells exhibited a small cuboidal epithelial phenotype (days 3 and 7, Figure 2C) with staining of PCK^+^/Vimentin^+^/Melan-A^−^ (day 10, Figure 2D), a characteristic feature of LEPC.

### 2.3. Characteristic Features of Sorted Cells

#### 2.3.1. CD90^+^CD117^−^ Cells (LMSC)

Flow cytometry analysis of LMSC (P1) revealed the expression (>95%) of CD44, CD73, CD90, CD105, and no expression (<0.5%) of CD11b, CD14, CD19, and CD45 (Figure 3A). LMSC cultures could be passaged 7 times with 55.0 ± 4.6 population doublings (PD) over 70–80 days and doubling time increased with passage number (Figure 3B). The proliferation potential (cell proliferation and growth rate) decreased with increasing passages (Figure 3B). A phase contrast micrograph illustrates a typical LMSC colony (Figure 3(Ci)) and a macroscopic image shows the crystal violet stained colonies in a T75 cm^2^ flask, when plated at 2 cells/cm^2^ (Figure 3(Cii)). LMSC in culture showed a colony-forming efficiency (CFE) of 56.0 ± 10.9% at passage 1 (P1), 64.2 ± 10.31% at P2, 52.2 ± 7.5% at P3, 41.8± 9.5% at P4, 18.7 ± 8.1% at P5, 9.4 ± 8.2% at P6, and 1.5 ± 1.6 at P7% (Figure 3(Ciii)). At P8, the cells showed no colony forming ability illustrating that the CFE decreases with increasing passages (Figure 3(Ciii)). LMSC were differentiated in vitro using adipogenic, osteogenic, and chondrogenic induction media. Three weeks after the adipogenic induction, the cells were successfully stained for fatty acid binding protein 4 (FABP4), which meant the cells were showing a lipid laden adipocyte phenotype (Figure 3D). Three weeks after osteogenic induction, the cells showed expression of osteocalcin and chondrogenically induced cells showed aggrecan expression (Figure 3D), suggesting their differentiation into the respective phenotypes. The undifferentiated (UD) controls displayed no FABP4 antibody labeling, whereas weak antibody labeling was observed for osteocalcin and aggrecan (Figure 3D).

#### 2.3.2. CD90^−^CD117^+^ Cells (LM)

To verify the phenotype of CD90^−^CD117^+^ enriched LM cell populations, established melanocytic markers were studied by immunocytochemistry. For immunostaining, LM (P1) were cultured on 4-well chamber slides in the presence of LN-511-E8 as a substrate. Immunostaining confirmed the expression of Melan-A, SRY-box transcription factor 10 (Sox10), human melanoma black-45 (HMB-45), and tyrosinase-related protein 1 (TRP1) (green) in all cultured CD90^−^CD117^+^ cells (Figure 4A). The self-renewal potential of LMs was evaluated by seeding cells at low density (20 cells/cm^2^) and the CFE was 81.0 ± 34.0% (Figure 4B). L-3,4-dihydroxyphenylalanine (L-DOPA) stimulated melanin production by cultivated LM (P1), which was indicated by macroscopic darkening of the culture medium (Figure 4B). The spectroscopic analysis of L-DOPA stimulated culture medium showed a five-fold increase in absorption compared to control medium (Figure 4B). These data show that enriched CD90^−^/CD117^+^ LM are functional in producing and secreting melanin into the culture medium.

#### 2.3.3. CD90^−^CD117^−^ Cells (LEPC)

Double immunostaining of LEPC (P1) indicated the expression of E-Cadherin (red), P(placental)-cadherin(green), CK14(green) in all cells; CK15 (green) and CK19 (red) in few cells (−10–20%); and CK3^+^ (green) cells were rarely seen (−1–2%) (Figure 4D). The majority of LEPC expressed the proliferation marker Ki-67 (red). After 10 days of culture, the total limbal population of LEPC showed minimal fibroblast contamination (Figure 4(Ei), arrow heads) and rarely melanocyte-like cells (Figure 4(Eii), arrow head), whereas no fibroblast or melanocyte-like cells were seen in the CD90^−^CD117^−^ LEPC population (Figure 4(Eiii)). The colony forming ability of CD90^−^CD117^−^ cells compared with that of total limbal cells after plating on mitomycin-c treated fibroblasts. No significant difference was observed between the samples in colony forming efficiency (1.4% of total limbal population vs. 1.5% of CD90^−^CD117^−^) or growth area (67.0% of total limbal population vs. 60.0% of CD90^−^CD117^−^) (Figure 4F).

#### 2.3.4. The 3D Co-Cultures

To mimic in vivo limbal niche interactions, a 3D co-culture was established using all three cell types. Phase contrast micrographs showed a confluent epithelial layer in both LEPC and LEPC-LM-LMSC transwell cultures (Figure 5A). The whole-mount immunostaining analysis showed expression of E-cadherin and vimentin in epithelial cells of both constructs (Figure 5B). In LEPC-LM-LMSC constructs, melanocytes were interspersed within the epithelial layers (Melan-A^+^(red)/vimentin^+^(cyan)) and vimentin^+^ LMSCs (Figure 5B).

Light microscopic analyses of tissue-engineered epithelial constructs showed multilayered cell sheets consisting of a cuboidal basal layer and 3–5 layers of suprabasal cells when co-cultured with LMs and LMSC, but only 2–3 cell layers in the absence of these niche cells, after 10 days of air-lifting (Figure 5C). Immunohistochemical analyses revealed the expression of epithelial keratins (PCK) in the epithelial cells on both types of construct (Figure 5D). The expression of vimentin was also observed in the basal layers of epithelium in both systems (Figure 5D, dotted line separates basal and suprabasal epithelium) and in stromal cells on another side of the insert of the LEPC-LM-LMSC construct (Figure 5D, arrow heads). The expression of CK12, a corneal-specific differentiation marker, was observed in all cells of LEPC constructs, but only in few superficial cells of LEPC-LM-LMSC constructs (Figure 5D, arrow heads). Expression of the LEPC marker CK14 was observed in the basal and transient amplifying cells, whereas CK19 was restricted to basal epithelial cells in both systems. p63^+^ cells were observed in both types of construct, but the number of p63^+^ cells was higher in the presence of LM and LMSC (Figure 5D, arrow heads). Moreover, we also observed Ki-67^+^ cells in the basal layer of the epithelium in LEPC-LM-LMSC constructs (Figure 5D, arrow heads). No Ki-67^+^ cells were observed in the LEPC constructs (Figure 5D).

## 3. Discussion

LEPC are located in a specialized microenvironment composed of ECM, limbal vasculature, and LNC. As native components of the limbal stem cell niche, LNC have the unique capability to determine the fate of LEPC both in vitro and in vivo [5,7,8,9]. However, the isolation of these LNC has proven to be difficult due the small fraction of LNC in the total limbal population and primary cultures were always hampered by contamination with other cell types [18,23]. In this study, we evaluated a new protocol to optimize the differential isolation of pure populations of LEPC as well as limbal niche cells for understanding the functional role of niche cells at limbal stem cell niche for future clinical applications. Earlier studies have shown that clusters derived from collagenase digestion included more epithelial progenitor cells, LMSCs, and melanocytes than dispase-isolated cell sheets [3,15]. Moreover, it has also been reported that cluster-derived cell suspensions contained more CD117^+^/Melan-A^+^ cells [24] and that stromal cells located immediately subjacent to limbal basal epithelial cells support LEPC better than stromal cells located in deeper stromal layers [18,19]. Hence, in the present study, we used the cluster-derived cells for efficient individual cell type-specific isolation of LEPC, LM and LMSC.

Various methods have been reported to isolate and expand human LMSCs either by explant culture of limbal tissue [21,31], or digestion of limbal tissue either by collagenase [15,17], dispase [19,20], or a combination of both [16,18]. It has been reported that limbal tissue treated with a combination of dispase and collagenase yielded clusters composed of approximately 95% mesenchymal cells and 5% epithelial cells whereas collagenase digestion of limbal tissue provided clusters consisting of roughly 20% mesenchymal cells and 80% epithelial cells [18]. However, all these methods did result in contamination of LMSC populations by epithelial cells (5–80%). In the current study, we used the combination of enzymatic digestion (using collagenase and trypsin) and flow sorting (using CD90 and CD117) to isolate LMSC. The sorted CD90^+^CD117^−^ cells exhibited fibroblastic morphology with expression of vimentin and a clear lack of epithelial keratins (PCK) and Melan-A, strongly suggesting a pure population of stromal cells. The isolated CD90^+^ cells fulfilled the criteria of MSC with characteristics of (a) plastic adherence, (b) a phenotypic profile of MSC (CD44^+^CD73^+^CD90^+^CD105^+^/CD11b^−^CD14^−^CD19^−^CD45^−^), and (c) multipotency (differentiation into adipocytes, osteocytes and chondrocytes), similar to earlier publications [18,21,32]. It has been reported that collagenase digestion-derived MSC could be expanded on matrigel for up to 12 passages with 33 cell doublings, whereas LMSC derived by dispase/collagenase digestion were expanded on matrigel for up to 10 passages with −25 population doublings [18,33]. MSC derived from limbal explant cultures could be propagated on plastic for up to six passages with 22.95 population doublings [21]. In the current study, sorted CD90^+^ LMSC presented superior proliferation capacity with 55.0 ± 4.6 population doublings over 70–80 days of culturing on plastic. We also evaluated the colony-forming capacity to evaluate MSC function [34]. LMSC showed more than 50% efficiency until P3, which is also superior to reports on dispase/collagenase-treated LMSC (−30%) [18] or explant culture-derived LMSC (30–40% (P2), 10–15% (P3)) [21]. These results clearly suggest that the protocol combining enzymatic treatment and sorting yielded a rather pure population of LMSC with superior self-renewal capacity compared to existing protocols.

Recently, we have reported on a novel protocol to obtain LM in a very short period of time and to avoid any toxic effects exerted by commonly used selection agents [24]. However, the sorted CD117^+^ LM populations still contained small amounts of contaminating stromal cells in most of the cultures (6/7 cultures) and required a second CD117-based FACS sort [24]. The refined strategy of using CD117 in combination with CD90 provided primary cultures of CD117^+^/CD90^−^ cells which did not show any fibroblast-like or vimentin^+^ melan-A^−^ cells (Figure 2C), suggesting a pure population of LM. In line with our earlier observations, the sorted LM have unabated proliferative potential (data not shown) and are functional in producing and secreting melanin [24]. Moreover, when seeded at low density, LM showed colony forming capability suggesting a melanocyte progenitor phenotype. Intriguingly, studies on epidermal melanocytes are also hampered by a lack of efficient protocols to isolate melanocytes. Traditional protocols require several weeks for epidermal melanocyte purification [35,36]. Recently, Willemsen et al. [37] reported a sorting protocol to isolate a population of epidermal melanocytes (CD45^−^CD3^−^HLA-DR^−^CD117^+^) within several hours. The protocol described in this article may also be used as an alternative method to receive a pure population of epidermal melanocytes and niche fibroblasts.

LEPC are a major component of the limbal niche and responsible for homeostasis of the corneal epithelium. In the past, LEPC have been isolated either by enzymatic digestion (dispase or collagenase) or explant culture of limbal tissue followed by enrichment with epithelial specific media in the presence or absence of feeder layers, which reduce fibroblast/stromal contamination [3,31,38,39]. It has been reported that cytospin preparations derived from collagenase-isolated clusters revealed more PCK^−^/Vimentin^+^ niche cells (mainly stromal cells and melanocytes) (19.5 ± 4.0%) than clusters derived from dispase (3.6 ± 2.2%) [15]. However, collagenase-isolated clusters included more epithelial progenitors (along with stromal cells) than dispase isolated sheets. This finding was supported by higher clonal growth capacity with a significantly higher number of holoclones and meroclones on 3T3 feeder layers [15]. Previous data suggest that the LEPC cultures used to carry a minimal fibroblast contamination [40,41] which was reduced by differential trypsinization to get pure populations [42]. In contrast, sorted CD90^−^/CD117^−^ cultures did not show any contamination by fibroblasts or melanocytes and expressed the progenitor-specific markers CK14, CK15, CK19, and Ki-67, suggesting a pure LEPC population. Moreover, the sorted LEPC showed CFE and colony growth area similar to the total limbal cell population, suggesting that the sorting protocol only removed LMs and LMSC, which are present in very low numbers, without disturbing the proliferation potential of LEPC. These observations strongly suggest the successful isolation of total limbal epithelial cell population without contamination of fibroblasts or melanocytes. We also observed that the number of sorted CD90^+^CD117^−^, CD90^−^CD117^+^ or CD90^−^CD117^−^ per limbal tissue varied from sample to sample (Figure 2B), but the percentage of CD90^+^CD117^−^ and CD90^−^CD117^+^ in the total limbal population remained the same. The variation in cell number is most likely due to donor age, tissue quality, and duration of organ culture (Appendix A).

Both LMSC and LM have been shown to support corneal epithelial regeneration during wound healing as well as maintenance of the LEPC phenotype both in vitro and in vivo [5,8,9,43]. Colony forming assays and 3D-Matrigel, -fibrin, or -transwell co-cultures have been used to investigate the potential of LNCs to maintain LEPC phenotypic status in vitro [9,22,23,44]. Recently, we reported that 2D co-cultures of LEPC with LMSC and LM as well as 3D co-cultures of LEPC and LM preserved the LEPC stem cell phenotype better than LEPC co-cultured with 3T3 fibroblasts or LEPC alone in transwells [9]. These observations suggest that direct LM contact and a close association with LMSC (paracrine) are involved in suppression of LEPC differentiation and stimulation of K15 and ABCG2 expression. In the current study, we evaluated a novel 3D culture method in which LEPC were mixed with LMs and physically separated from LMSC by a fluid-permeable membrane, to mimic the in vivo limbal stem cell niche. After two weeks of cultivation, the data showed superior growth capacity, stratification, and preservation of a stem cell phenotype (more p63^+^ and Ki-67^+^ cells) as compared to LEPC monocultures. These results are similar to observations made in 3D-fibrin LEPC-LM co-cultures [23] and 3D-transwell co-cultures of LEPC with either LMSC or LM [9,44]. These results strongly suggest that LEPC, LMSC, and LMs may act in concert both in native limbal niche as well as tissue-engineered limbal epithelial carrier constructs. Moreover, co-cultivation of LEPC with LM/LMSC could represent an improved strategy to better maintain LEPC stem cell phenotype and improve long-term results of cultivated limbal epithelial cell transplantation in patients suffering from limbal stem cell deficiency. However, further studies are warranted to elucidate the nature of signaling pathways activated by the limbal niche cell–cell interactions (direct contact or paracrine) determining the fate of LEPC.

The major benefit of our protocol is that pure and functional LEPC, LMSC, and LM can be obtained within several hours (−1 day) from single preparation, allowing direct transcriptome analysis or proteomic profiling and functional studies on native unpassaged LNC. The biomimetic co-culture method presented in this article provides an experimental model for investigating the functional role of LM and LMSC in the limbal stem cell niche, the pathological conditions generated at the limbus, and their suitability for developing advanced therapies.

## 4. Materials and Methods

Human donor corneoscleral tissues with appropriate research consent was provided by the Lions Cornea Bank Baden-Württemberg after retrieval of corneal endothelial transplants as described previously [24]. Informed consent to corneal tissue donation had been obtained from the donors or their relatives. Experiments using human tissue samples were approved by the Institutional Review Board of the Medical Faculty of the University of Freiburg (25/20) and adhered to the tenets of the Declaration of Helsinki.

### 4.1. Cell Isolation

Limbal cells were isolated as previously described [42]. Briefly, organ-cultured corneoscleral tissue (*n* = 115, mean age 69.8 ± 10.7 yrs; culture duration 24.0 ± 4.9 days; post-mortem time 33.54 ± 17.4 h; light pigmented donor limbal tissue; Appendix A) was cut into 12 three-clock-hour sectors, from which limbal segments were obtained by incisions made at 1 mm central to and peripheral of the anatomical limbus. Limbal segments were enzymatically digested with collagenase A (Sigma-Aldrich, St. Louis, MO, USA; 2 mg/mL) at 37 °C for 18 h to generate cell clusters containing mixtures of epithelial, mesenchymal, and melanocytic cells. Cell clusters were separated from single cells by using reversible cell strainers with a pore size of 37 µm (Stem Cell Technologies, Köln, Germany). Subsequently, the cell clusters were dissociated into single cells using 0.25% Trypsin-EDTA at 37 °C for 10–15 min. The obtained single cells from pooled corneoscleral tissues (4–6 corneae in a single preparation) were further processed for sorting as described below.

Limbal cell clusters were also cultured in 4-well chambers for 10 days in corneal culture medium (CCM) containing Dulbecco’s modified Eagle medium/Ham’s F12 (3:1) (Hyclone; GE Healthcare Life Sciences, Freiburg, Germany) supplemented with bovine pituitary extract (BPE, 25 µg/mL), epidermal growth factor (EGF, 0.15 ng/mL) (Life Technologies, Carlsbad, CA, USA), 5% fetal calf serum (GE Healthcare Life Sciences), penicillin (100 U/mL)-streptomycin (100 µg/mL) mix (Sigma-Aldrich) and processed for immunocytochemistry as described below.

### 4.2. Fluorescence-Activated Cell Sorting (FACS)

FACS was carried out as described previously [24]. Briefly, single-cell suspensions were incubated with FcR blocking reagent (Miltenyi Biotec, Bergisch Gladbach, Germany; 20μL/10^6^ cells) for 5 min. Subsequently, cells were washed and incubated with mouse anti-human CD117-PE and CD90-APC (5 μL/10^6^ cells) (ebiosciences, San Diego, CA, USA) in 100 µL phosphate-buffered saline (PBS, Boston, MA, USA), 0.1% sodium azide, and 2% fetal calf serum for 30 min at 4 °C in the dark. Cells were then washed and DAPI (1:5000) was added to exclude dead cells. The sorting was performed using a FACS Aria II sorter (BD Biosciences, Heidelberg, Germany) and the FACSDiva software (BD Pharmingen; BD Biosciences). Post-acquisition analysis was performed using FlowJo software (Tree Star, Inc., Ashland, OR, USA).

After sorting, CD117^+^/CD90^−^ cells (LM) were seeded in LN-511-E8- (iMatrix-511, Nippo; 0.5 µg/cm^2^) coated 12-well plates (Corning, Tewksbury, MA, USA) and cultured in CNT-40 medium (CellnTec, Bern, Switzerland). The CD90^+^CD117^−^ (LMSC) cells were seeded in 12-well plates (Corning, Tewksbury, MA, USA) and cultured in Mesencult media (Stem Cell Technologies). CD117^−^/CD90^−^ cells (epithelial fraction, LEPC) were seeded on 3T3 fibroblasts for colony forming assays (described below) or seeded into T75 flasks in Keratinocyte serum-free medium (KSFM) supplemented with BPE (25 µg/mL) and EGF (0.15 ng/mL) (Life Technologies) for expansion. All cultures were maintained at 37 °C, 5% CO_2_, and 95% humidity and media changed every other day.

### 4.3. Flow Cytometry

Flow cytometry was carried out as previously described [24]. Briefly, single-cell suspensions (0.5–1 × 10^6^ cells) were incubated with Fluorochrome-conjugated antibodies and respective isotype controls (Appendix A). After three washes, cells were resuspended in ice-cold PBS, and flow cytometry was performed on a FACSCanto II (BD Biosciences) by using FACS Diva Software as described above.

### 4.4. Growth Characteristics

#### 4.4.1. Population Doubling Assay

A population doubling (PD) and proliferation assay was performed as described previously [21]. The assay was performed on LMSC from passage 0 until no further cell growth after passaging was seen. Cells were plated at a given density (1 × 10^4^ cells/T75 flask) with each passage and trypsinized after 10 days. The number of cells was determined by using a Neubauer counting chamber. The population doubling of cells was calculated as:The number of cell doublings (NCD) = log_10_(y/x)/log10_2_(1)
where y is the final density of the cells and x is the initial seeding density of the cells. The cumulative population doublings are the sum of PDs in all passages.

Doubling time is calculated from the cell number and the time of cell counting, using the following formula:Doubling time = (t − t0)log2/(logy − logx),(2)
where t, t_0_ represents the time at cell counting; y equals the number of cells at time t, and x equals the number of cells at time t_0_.

Growth rate is calculated from the initial and final cell number of each passage and number of days in culture, using the formula:Growth rate: ln(N_t_/N_o_)/t(3)
where N_t_ represents final cell number; N_o_ represents the initial cell number and t equals number of days in culture.

#### 4.4.2. Colony-Forming Unit Assay of LMSC and LM

Clonal Expansion of LMSC and LM was performed as described earlier [21]. Briefly, LMSC (P0-P6, two cells per cm^2^) and LM (P1, 20 cells/cm^2^) were cultured (T75 cm^2^ flasks) in respective media as described above for 14 and 20 days, respectively. The cultures were stained with 0.5% crystal violet in methanol for 5 min. The colony count was performed and colonies that were less than 2 mm in diameter or faintly stained were excluded. The colony-forming efficiency (CFE) was calculated using the formula: number of colonies formed/ number of cells plated × 100%.

### 4.5. Trilineage Differentiation

Adipogenic, osteogenic, and chondrogenic differentiation assays on LMSC (P1) were carried out using a Human MSC functional identification kit (SC006, R&D systems; Wiesbaden, Germany). For adipogenic differentiation, MSCs were seeded into a 4-well chamber slide at a density of 3.5 × 10^4^ cells/well and maintained in culture medium until 100% confluency. Cells were then exposed to adipogenic differentiation medium for 3 weeks. For osteogenic differentiation, 7.4 × 10^3^ cells were seeded per well. When cells reached 50–70% confluency, the medium was replaced with osteogenic differentiation medium and kept for 3 weeks. Chondrogenic differentiation was tested as described earlier [45]. For chondrogenic differentiation, 5.0 × 10^4^ cells were placed in 4-well chamber slides and maintained in chondrogenic differentiation medium for 3 weeks. After incubation, differentiation potential of LMSC was assessed by immunostaining using primary antibodies raised against fatty acid binding protein 4 (FABP4, adipogenic), osteocalcin (osteogenic), or aggrecan (chondrogenic).

### 4.6. Melanin Production

Melanin production was assessed as described earlier [24]. Briefly, cells were seeded at 1 × 10^5^ cells per well in CNT-40 medium in a 12-well plate and cultured for 24 h at 37 °C in the absence or presence of 1 mM L-DOPA to stimulate melanin synthesis. After 24 h, the culture medium was collected. Next, 100 µL of 1 M sodium hydroxide was added to 100 µL culture medium to dissolve melanin at 70 °C for 90 min. Melanin concentration was determined by comparing 405 nm absorbance values in a Spark microplate reader (TECAN) from experimental samples with a standard curve ranging from 0 to 100 µg/mL generated with synthetic melanin (Sigma). Synthetic melanin was dissolved using 1 M sodium hydroxide solution in water. The fold change values were calculated as OD of the Induction/OD of control (*n* = 5).

### 4.7. Co-Culture Experiments

#### 4.7.1. Colony-Forming Unit Assay of LEPC

Clonal expansion of LEPC was studied on feeder layers using mitomycin C-treated 3T3 fibroblast as described previously [9]. The sorted live cells from isotype controls (total limbal population) and the CD90^−^/CD117^−^ population (as mentioned above) were seeded at a density of 300 cells/cm^2^ on the feeder layer. After 14 days of culture in CCM, the colonies were stained using 0.5% crystal violet. The CFE was calculated as described above and the colony growth area was calculated as colony growth area/total culture area × 100%. For colony counting, holoclones, meroclones, and paraclones were included in the counting.

#### 4.7.2. The 3D Co-Cultures

The LEPC (P1) were co-cultured with a combination of mitomycin C-treated LMSC (5 µg mitomycin-C/mL medium for 2 h) and active LMs (LEPC-LMSC-LM). LEPC cultures served as controls. For LEPC-LMSC-LM constructs, mitomycin-treated LMSC (2.5 × 10^4^ cells/insert) were seeded on the backside of 12-well inserts (BRANDplates^®^, 1 μm pore size, PC-membrane; BRAND GmbH, Wertheim, Germany) and incubated at 37 °C overnight to allow for cell attachment as described previously [41]. Subsequently, LEPC (7.5 × 10^4^/insert) and LM (2.5 × 10^4^/insert) were seeded (3:1) on the upper side of the membrane. For LEPC controls, LEPC were seeded on top of the membrane and cultured in CCM media. After LEPC confluence, the cells were raised to the air–liquid interface and cultured for 10–12 days. All cultures were maintained at 37 °C, 5% CO_2_, and 95% humidity and medium was changed every other day. For final evaluation, the inserts were fixed for immunohistochemistry and light microscopy as described below.

### 4.8. Histology and Immunohistochemistry—Paraffin

For routine histology, 3D-sandwich culture inserts were fixed in 4% paraformaldehyde (30 min) and embedded in paraffin. The 5 µm thick sections were cut and stained as described previously [9]. Briefly, sections were stained with hematoxylin (Haematoxylin Gill III, Surgipath, Leica, Germany) for 2 min and 1% eosin Y (Surgipath, Leica, Germany) for 1 min to examine the gross architecture and epithelial stratification.

Immunostaining of paraffin sections of 3D-sandwich culture inserts was performed as previously described [46]. The list of antibodies is provided in Appendix A.

### 4.9. Immunohistochemistry—Frozen and Immunocytochemistry

Corneoscleral tissue samples (mean age 75.2 ± 10.9 yrs) within 16 h after death were embedded in optimal cutting temperature (OCT) compound and frozen in liquid nitrogen. Cryosections of 6 μm thickness were cut from the superior or inferior quadrants and cells cultured on 4-well glass chamber slides (LabTek; Nunc, Wiesbaden, Germany) were fixed in 4% paraformaldehyde for 15 min, blocked with 10% normal goat serum (NGS), and incubated in primary antibodies (Appendix A) diluted in 2% NGS, 0.1% Triton X-100 in PBS overnight at 4 °C or 3 h at room temperature. Antibody binding was detected by Alexa-488-, -555-, -647-conjugated secondary antibodies (Life Technologies, Carlsbad, CA, USA) and mounted in Vectashield antifade mounting media with DAPI (Vector, Burlingame, CA, USA). Immunolabeled cryosections and cultured LM were examined with a laser scanning confocal microscope (TCS SP-8, Leica, Wetzlar, Germany). For negative controls, the primary antibodies were replaced by PBS. For wholemount assays, 3D co-culture inserts were fixed in 4% paraformaldehyde for 20 min and immunostaining was carried out as described above.

### 4.10. Statistical Analysis

The statistical analyses were performed as described earlier [24]. Briefly, the GraphPad InStat statistical package for Windows (Version 6.0; Graphpad Software Inc., La Jolla, CA, USA) was used to perform statistical analyses. Results are expressed as mean ± standard deviation (SD) from individual experiments or as mean ± standard error of the mean (SEM) (graphs). The statistical significance (*p* value < 0.05) was determined with the Mann–Whitney U test.

## Figures and Tables

**Figure 1 ijms-23-02750-f001:**
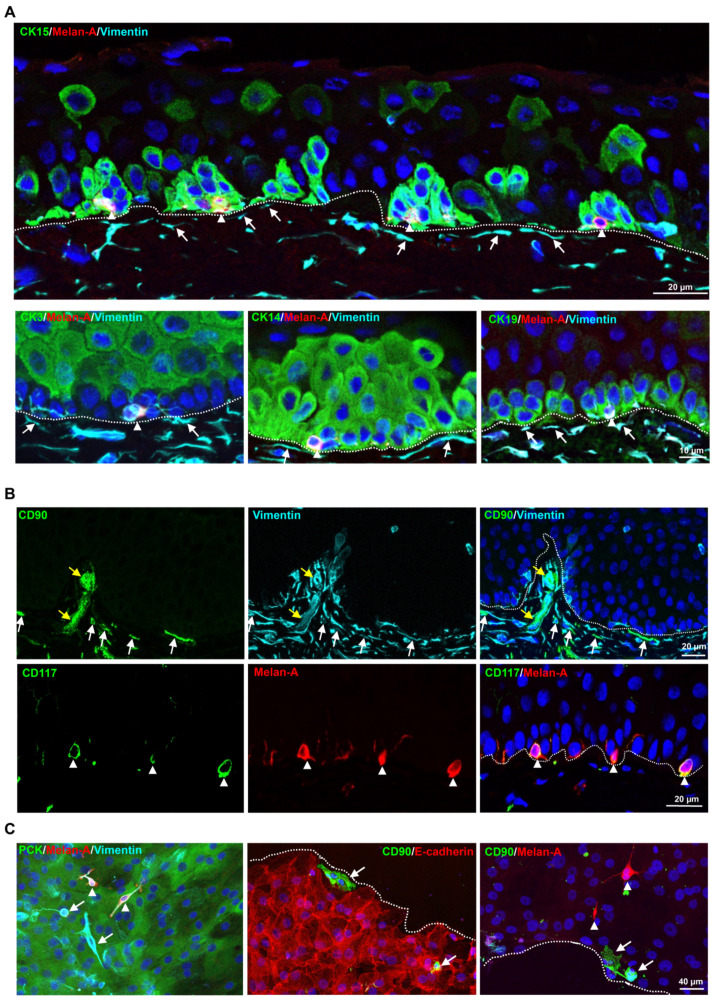
Localization of limbal niche cells in situ: (**A**) triple immunostaining analysis of limbal tissue sections showing the melanocytes (Melan-A^+^ (red) vimentin^+^ (cyan) cells, arrow heads) close contact with clusters of cytokeratin (CK)15^+^; CK14^+^, CK19^+^ (green) limbal epithelial progenitor cells (LEPC), whereas sub-epithelial stromal cells (vimentin^+^ cells (cyan), arrows) were in close association with basal limbal epithelial cells and not with more superficial CK3^+^ cells (green) (dashed line represents the basement membrane (BM)). Nuclear counterstaining with 4′,6-diamidino-2-phenylindole (DAPI, blue). (**B**) Double immunostaining of limbal sections showing the co-localization of CD90 (green) and vimentin (cyan) in the sub-epithelial stromal cells (white arrows), which were in close association with basal layers of limbal epithelium (dotted line represents the BM) as well as blood vessels of the limbal stroma (yellow arrows). The limbal sections also showing the co-localization of CD117 (green) and Melan A (red) in the melanocytes (arrow heads) at the basal layer of limbal epithelium. Nuclear counterstaining with DAPI (blue). (**C**) Immunofluorescence analysis of cultured limbal clusters showing the expression of keratins (PCK, green) and vimentin (cyan) in epithelial cells; Melan-A (red) and vimentin (cyan) expression in melanocytes (arrow heads) and only vimentin expression in stromal cells (arrows). Double immunostaining of cultured limbal clusters showing the CD90^+^ stromal cells (green, arrow) at the edge of clusters and also within the E-cadherin^+^ epithelial cells (red, dashed line represents the edge of the cluster), whereas Melan-A^+^ melanocytes between the cells (red, arrow heads). Nuclear counterstaining with DAPI (blue).

**Figure 2 ijms-23-02750-f002:**
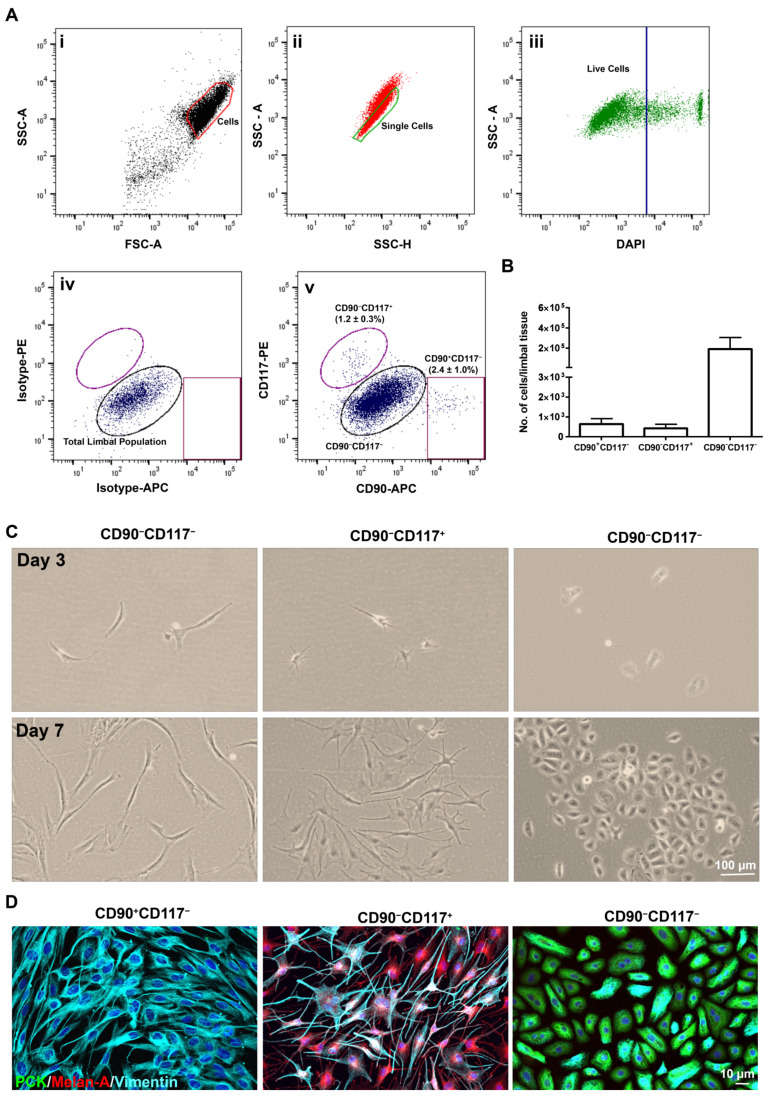
Flow sorting of limbal niche cells and characterization: (**A**) fluorescence activated cells sorting (FACS) images demonstrating the gating strategy used to isolate limbal niche cells. Forward scatter (FSC-A) vs. side scatter (SSC-A) graph showing the selected cells of interest based on size and granularity (i). Side scatter area vs. width graph showing the selection of single cells by excluding doublets or clumps, (ii) followed by dead cell exclusion using 4′,6-diamidino-2-phenylindole DAPI (iii). The isotype control graph showing the set of gates (iv) to select the cells of CD90^+^CD117^−^, CD90^−^CD117^+^, and CD90^−^CD117^−^ cells (iv). Percentages (%) of positive cells are expressed as the means ± SEM of 23 individual experiments. (**B**) The graph showing the percentage of CD90^+^CD117^−^, CD90^−^CD117^+^, CD90^−^CD117^−^ cells obtained from limbus. Data are expressed as the means ± SEM of 23 individual experiments including 115 corneoscleral tissues. (**C**) Phase contrast images showing the spindle-shaped morphology, elongated with prominent nucleolus of CD90^+^CD117^−^ cells; large, flattened, smooth bodies with multiple dendrites of CD90^−^CD117^+^ cells; small cuboidal epithelial phenotype of CD90^−^CD117^−^ cells. (**D**) Triple immunostaining analysis of cultured cells showing the vimentin^+^ cells in CD90^+^CD117^−^ cell cultures; Melan-A (red) and vimentin (cyan) double positive cells in CD90^−^CD117^+^ cultures; CD90^−^CD117^−^ cells stained for pan-cytokeratin (PCK) and vimentin. Nuclear counterstaining with 4′,6-diamidino-2-phenylindole (blue).

**Figure 3 ijms-23-02750-f003:**
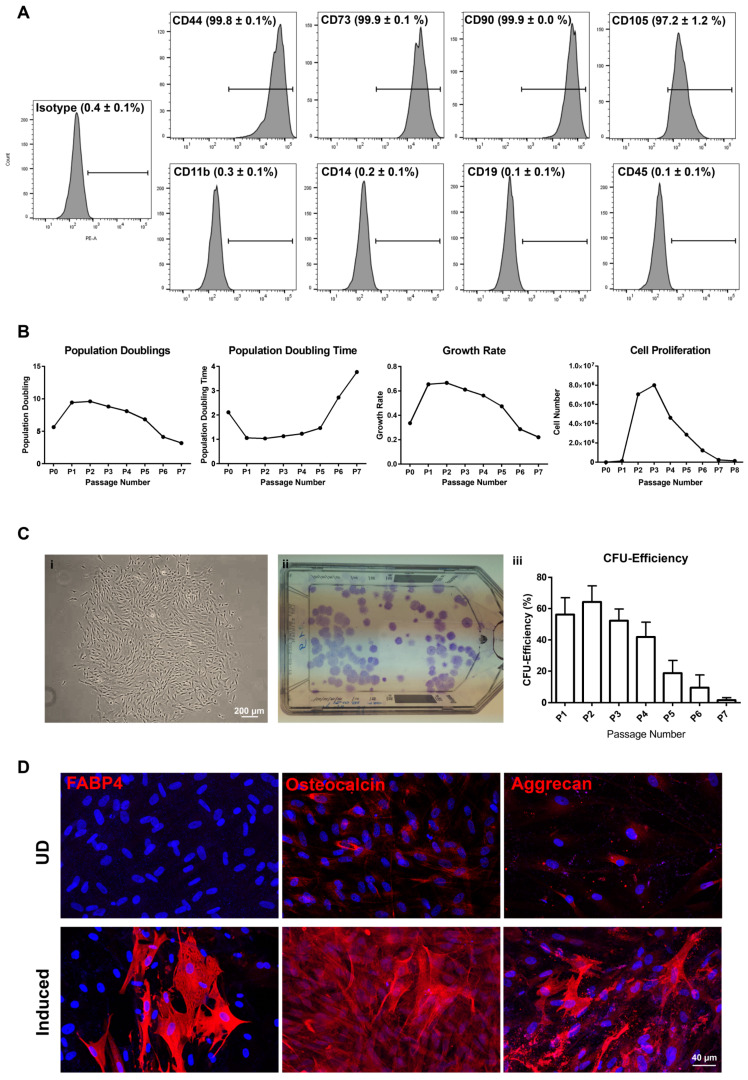
Phenotypic profile and functional characterization of CD90^+^CD117^−^ (LMSC) cells: (**A**) flow cytometry analysis showing the expression of CD markers. Percentage of cells expressed mean ± SEM of 4 individual experiments. (**B**) Graphs showing the population doublings, population doubling time, growth rate, and proliferation potential of LMSC over the passages. Data are expressed as means of 5 individual experiments. (**C**) Phase contrast micrograph showing the LMSC colony (i) and T75 flask showing crystal violet stained colonies of LMSC (ii). The graph represents the colony forming efficiency of LMSC over the passages. Percentage of colonies expressed as means ± standard deviation (*n* = 5). (**D**) Immunostaining analysis showing the expression of fatty acid binding protein 4 (FABP4), osteocalcin and aggrecan in adipogenic, osteogenic, and chondrogenic induced cells, respectively. No staining has been seen for FABP4 in undifferentiated (UD) controls, but weak staining observed for osteocalcin and aggrecan in UD controls. Nuclear counterstaining with 4′,6-diamidino-2-phenylindole (blue).

**Figure 4 ijms-23-02750-f004:**
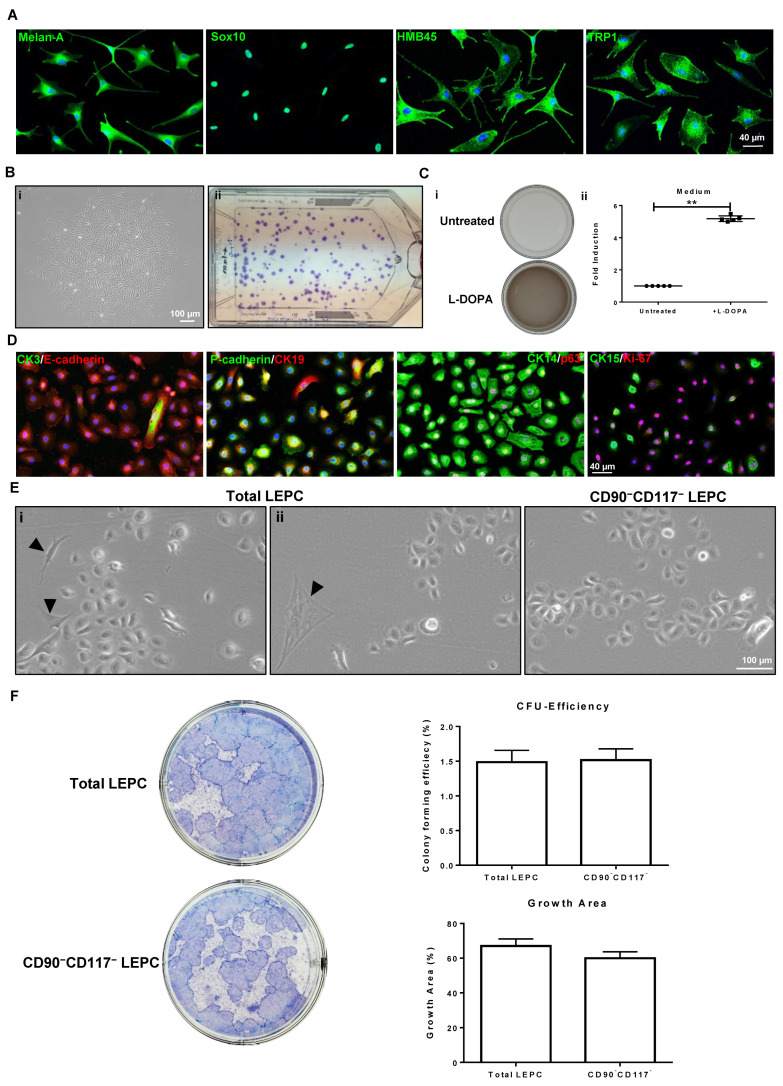
Phenotypic profile and functional characterization of CD90^−^CD117^+^ limbal melanocytes (LM) and CD90^−^CD117^−^ limbal epithelial progenitor cells (LEPC): (**A**) immunocytochemical analysis of cultured CD90^−^CD117^+^ cells showing expression of melanocyte markers Melan-A, sex-related HMG box 10 (Sox-10), tyrosinase-related protein 1 (TRP1 or TYRP1), and human melanoma black-45 (HMB45) (red); nuclear counterstaining with 4′,6-diamidino-2-phenylindole (DAPI; blue). (**B**) Phase contrast micrograph showing the LM colony (i) and T75 flask showing crystal violet stained colonies of LM (ii). (**C**) The cultured wells of LM in the presence or absence of 1 mM L-3,4-dihydroxyphenylalanine (L-DOPA) for 24 h showing light brown coloring of the culture medium, as can be observed macroscopically (i). The graph showing the l-DOPA stimulation significantly increased the melanin concentration in the medium to five-fold compared to unstimulated condition (iii). Data are expressed as means ±  SEM (*n*  =  5) ** *p*  <  0.01; Mann–Whitney *U* test. (**D**) Double immunostaining of cultured CD90^−^CD117^−^ cells showing the expression of epithelial (E)-Cadherin (red), placental (P)-cadherin (green), cytokeratin (CK)14 (green) in all cells; CK15 (green) and CK19 (red) in few cells (−10 to 20%); CK3^+^ (green) cells were rarely seen (−1 to 2%); Ki-67 expression in most of the cells. Majority of LEPC expressed the proliferative marker Ki-67 (red). Nuclear counterstaining with DAPI (blue). (**E**) Phase contrast micrographs of cultured cells showing the contamination of stromal cells (i, arrow heads) and melanocyte-like cells (ii, arrow heads) in total limbal population cultures; none of these cells were observed in CD90^−^CD117^−^ cell cultures. (**F**) Total limbal population LEPC and CD90^−^CD117^−^ LEPC form typical cellular colonies on the NIH/3T3 fibroblast feeder layers after 14 days in culture. Colony forming analysis showing no significant differences between the samples. Percentage of colony forming efficiency and growth area expressed as means ± standard error of the mean of 4 individual experiments.

**Figure 5 ijms-23-02750-f005:**
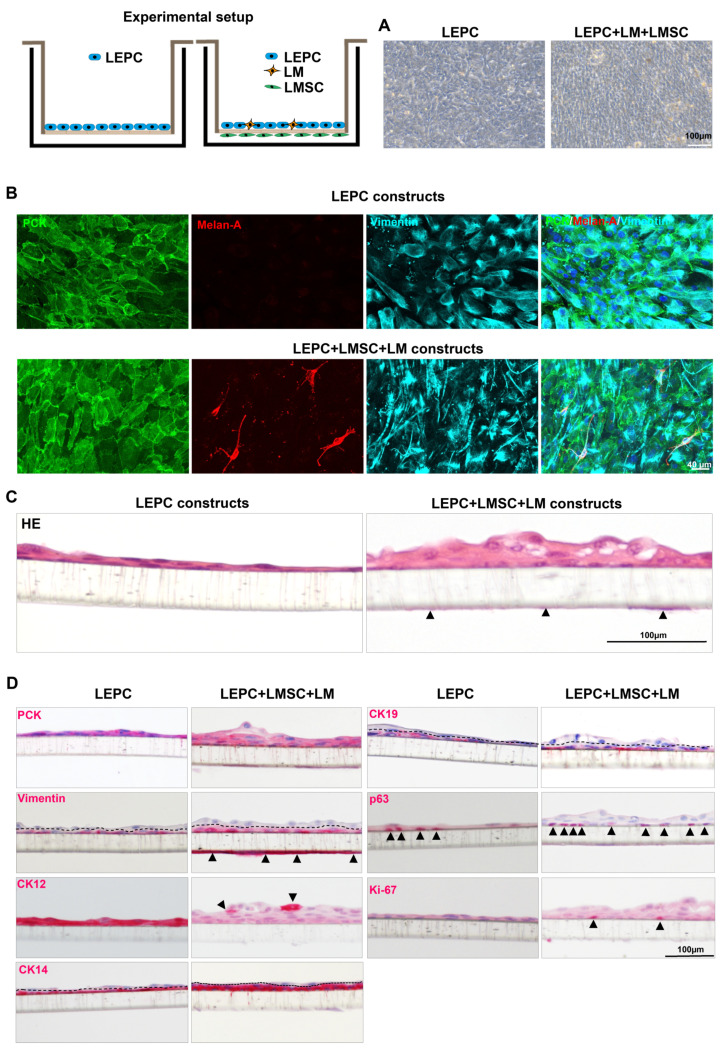
Effect of limbal mesenchymal stromal cells (LMSC) and limbal melanocytes (LM) on limbal epithelial progenitor cells (LEPC) in a 3D co-culture system: (**A**) phase contrast micrographs showing confluent epithelial layer in both LEPC and LEPC-LM-LMSC constructs. (**B**) The whole-mount triple immunostaining showing the expression of E (epithelial)-cadherin and vimentin in epithelial cells of both the constructs; vimentin^+^ (red) Melan-A^+^(cyan) melanocytes (interspersed within the epithelial cells) and vimentin^+^ LMSC in LEPC-LM-LMSC constructs. (**C**) Hematoxylin and Eosin (HE) staining of cell constructs showing multilayered cell sheet with 2–3 layers in LEPC constructs; 3–5 epithelial layers in LEPC-LMSC-LM constructs. Arrow heads indicating the LMSC. (**D**) Immunohistochemical analyses cell constructs showing the expression of the epithelial keratins (PCK) in the epithelial cells on both the constructs (Figure 5D); vimentin expression in the basal layers of epithelium in both constructs (dotted line separated basal and suprabasal epithelium) and in stromal cells on another side of the insert of LEPC-LM-LMSC construct (arrow heads); CK12 expression in all cell layers of LEPC constructs and few superficial cells in LEPC-LM-LMSC constructs (arrow heads); CK14 expression in the basal and transient amplifying cells; CK19 restricted to basal epithelial cells in both constructs; the p63^+^ cells (arrow heads) were observed in both constructs but the number of p63^+^ cells was higher in the LEPC-LM-LMSC constructs compared to LEPC constructs; Ki-67^+^ (arrow heads) cells observed in the basal layer of the epithelium in LEPC-LM-LMSC constructs, but none were observed in the LEPC constructs.

## Data Availability

The datasets generated during and/or analyzed during the current study are available from the corresponding author on reasonable request.

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
