# Peer review of "Efficient Isolation and Functional Characterization of Niche Cells from Human Corneal Limbus"

_ijms, 2022, doi:10.3390/ijms23052750_

Round 1

Reviewer 1 Report

Dear Authors,

The authors report the isolation of limbal epithelial progenitors’ cells, limbal mesenchymal stromal cells, and limbal melanocytes from donors, in order to characterize the single cells phenotype and in order to reconstruct a corneal epithelium. Limbal niche cells population was characterized with specific marker, in term of population doubling and growth. The authors reported that the population doubling, and CFU-efficiency decreased, when the number of cells passaging increases. Melanocytes limbal cells were also characterized with CK19, P-Cadherin markers and the CFU-Efficiency. To finish, the authors reconstituted a better corneal epithelium when limbal epithelial progenitors’ cells, limbal mesenchymal stromal cells, and limbal melanocytes were combined, compared to limbal epithelial progenitors’ cells alone. The cell sheet was more stratified when all the cells were combined. Those cell sheets could be used for corneal transplantation.

I have few comments

Major

  • For the figure 3 and 4, could the authors explain how they calculated the CFU? Did they count holoclones, meroclones and paraclones?
  • The 3D co-cultures were stopped from 10 to 12 days. What criteria did the authors used to decide the stop the 3D co-cultures?
  • Could the authors explain how they detect the proteins in the 3D co-culture (figure 5D), because it seems that the 3D co-culture is only dyed with H&E and I didn’t read the information in the materials and methods? Did they use HRP?

Minor:

  • All latin words must be in Italic.

Author Response

Response to Reviewer 1

We thank the reviewer for the suggestions to make the manuscript clearer. We believe we have addressed all issues raised

Point 1: For the figure 3 and 4, could the authors explain how they calculated the CFU? Did they count holoclones, meroclones and paraclones?

Response 1: The colony forming unit- efficiency was calculated using the formula: number of colonies formed/ number of cells plated ×100%. For LMSC and LM, the colonies that were less than 2 mm in diameter or faintly stained were excluded. The same has been mentioned in the methods section 4.4.2, page no. 15.

In LEPC, we tested the overall colony forming potential of CD117-CD90- vs total limbal epithelial cells. For this reason, we have included holoclones, meroclones and paraclones in the counting. For better clarity the following sentence has been added in the methods section 4.7.1, page no. 16.

“For colony counting of LEPC, holoclones, meroclones and paraclones were included in the counting”.

Point 2: The 3D co-cultures were stopped from 10 to 12 days. What criteria did the authors used to decide the stop the 3D co-cultures?

Response 2: The 3D co-cultures were stopped after 10 to 12 days of cultivation. It was based on our previous publication, where we have reported a multilayered epithelial sheet upon co-culture with limbal melanocytes, after 10 days of cultivation (Polisetti et al., 2020)*.

Point 3: Could the authors explain how they detect the proteins in the 3D co-culture (figure 5D), because it seems that the 3D co-culture is only dyed with H&E and I didn’t read the information in the materials and methods? Did they use HRP?

Response 3: We have done immunostaining of paraffin embedded 3D-sandwich culture inserts. The same has been mentioned in the second paragraph, methods section 4.8, page no. 17. However, for better clarity the sentence has been changed as follows…

“Immunostaining of paraffin sections of 3D-sandwich culture inserts was performed as previously described [46]”

Point 4: All latin words must be in Italic

Response 4: All latin words has been changed to Italic

*Polisetti, N., Gießl, A., Li, S. et al. Laminin-511-E8 promotes efficient in vitro expansion of human limbal melanocytes. Sci Rep 10, 11074 (2020).

Reviewer 2 Report

I red with great pleasure the manuscript. The text is written with an excellent English and clarity. The images and graphs very well illustrate the findings of the manuscript, which includes important new protocols and information for the scientific community.

I suggest publication of the manuscript following minor changes.

Page 10. line 3, please remove „our” once

Please include the exact Penicillin/Streptomycin concenteration for the cultures, in the methods (Page 13).

Author Response

Response to Reviewer 2

I red with great pleasure the manuscript. The text is written with an excellent English and clarity. The images and graphs very well illustrate the findings of the manuscript, which includes important new protocols and information for the scientific community.

We thank the reviewer for the kind appreciation of our work. We also thank the reviewer for the suggestions to improve the quality of the manuscript.

Point 1: Page 10. line 3, please remove „our” once

Response 1: The word „Our“ has been removed

Point 2: Please include the exact Penicillin/Streptomycin concenteration for the cultures, in the methods (Page 13).

Response 2: We have been added the following concentrations in the methods section - 100 U/ml Penicillin; 100 µg/ml Streptomycin

Round 2

Reviewer 1 Report

Dear authors,

Thanks for the answers. I have no additional questions.

I accept the manuscript as it is.